# UniTTS: Towards End-to-End Speech Synthesis with Joint Acoustic-Semantic Modeling

## Abstract

Recent advancements in multi-codebook neutral audio codecs, such as Residual Vector Quantization (RVQ) and Group Vector Quantization (GVQ), have significantly advanced text-to-speech (TTS) systems based on large language models (LLMs), whose exceptional capabilities in discrete token modeling have garnered significant attention within the speech processing community. However, since semantic and acoustic information cannot be fully aligned, a significant drawback of these methods when applied to LLM-based TTS is that large language models may have limited access to comprehensive audio information. To address this limitation, we propose DistilCodec and UniTTS, which collectively offer the following advantages: 1) DistilCodec distills a multi-codebook audio codec into a single-codebook codec with 32,768 codes, achieving near 100% codebook utilization. 2) By avoiding semantic alignment constraints, DistilCodec enables the incorporation of extensive high-quality unlabeled audio—such as audiobooks with sound effects and musical segments—during training, thereby enhancing data diversity and general applicability. 3) Leveraging the comprehensive audio information modeling of DistilCodec, we integrated three key tasks into UniTTS's pre-training framework: audio modality autoregression, text modality autoregression, and speech-text cross-modal autoregression. This allows UniTTS to accept interleaved text and speech/audio prompts while substantially preserving LLM's text capabilities. 4) UniTTS employs a three-stage training process: Pre-Training, Supervised Fine-Tuning (SFT), and Alignment. Experiments demonstrate that DistilCodec effectively resolves codebook collapse in large, single-codebook settings. Building on this, UnitTTS demonstrates remarkable capabilities for zero-shot voice cloning with emotional expression.

## 1 Introduction

In recent years, Large Language Models (LLMs) Radford et al. (2019); Kaplan et al. (2020); Grattafiori et al. (2024); Yang et al. (2024) have made remarkable progress, showing strong ability in modeling discrete tokens. This success has drawn growing interest from the speech processing community. At the same time, advances in multimodal discretization methods—such as Vector Quantization (VQ) Van Den Oord et al. (2017), Finite Scalar Quantization (FSQ) Mentzer et al. (2023), and Grouped-Residual-Factorized Vector Quantization (GRFVQ, including Grouped-VQ, Residual-VQ, and Factorized-VQ) Zeghidour et al. (2021); Yang et al. (2023); Yu et al. (2021)—have greatly improved Neural Audio Codecs (NAC). Building on these developments, many recent Text-to-Speech (TTS) systems have started to adopt LLM-based approaches Du et al. (2024b;a); Wang et al. (2025); Ye et al. (2025b); Liao et al. (2024); Deng et al. (2025); Chen et al. (2024); Anastassiou et al. (2024), achieving notable gains in both speech naturalness and emotional expressiveness.

The performance of LLM-based text-to-speech (TTS) systems strongly depends on the discrete audio tokens produced by NACs Li et al. (2024b); Ye et al. (2025a). Recently, most NACs have adopted semantic distillation Zhang et al. (2023); Défossez et al. (2024); Ye et al. (2025a;b); Wang et al. (2025) to enrich token representations with high-level information from audio encoders Baevski et al. (2020); Radford et al. (2023), enabling more expressive LLM-based TTS. However, this paradigm remains fundamentally limited: not all aspects of speech can be factorized into independent semantic and acoustic features, particularly in prosodically salient non-linguistic vocalizations (e.g., laughter, crying) and in high-fidelity universal audio. Moreover, attempts at joint modeling

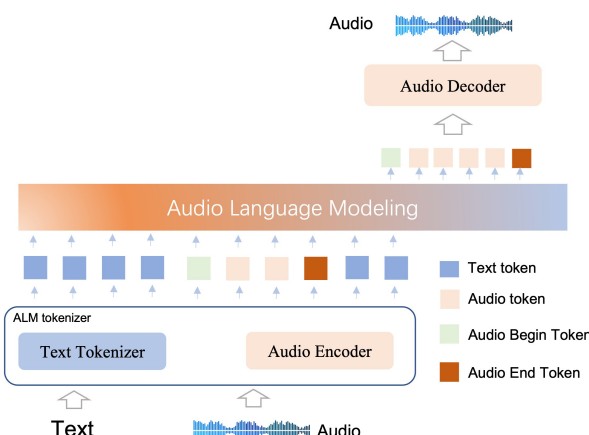

Figure 1: The UniTTS architecture consists of an ALM tokenizer and an ALM backbone network, supporting both text and audio inputs and outputs. Within the architecture, DistilCodec is responsible for audio signal transformation: its encode module discretizes audio into latent representations, while the decode module reconstructs the waveform for acoustic output.

of semantic and acoustic information often cause severe codebook collapse, which in turn degrades audio reconstruction quality. While GRFVQ-based multi-codebook NACs improve fidelity, they also inflate bitrates, making LLM sequence modeling substantially more difficult Li et al. (2024a); Xin et al. (2024); Parker et al. (2024); Wang et al. (2025). Compressed quantizers are effective at reducing the bitrate, but codebook utilization diminishes as the codebook size grows within a single-codebook architecture Ji et al. (2024). These limitations highlight the need for new approaches that achieve low-bitrate, high-fidelity universal audio representations while maintaining effective compatibility with LLM-based generation.

To address these limitations, we first introduce DistilCodec, a universal single-codebook audio codec, and then build UniTTS upon it. DistilCodec compresses a multi-codebook NAC into a single large-vocabulary codebook with 32,768 codes, achieving nearly 100% utilization and balanced distribution. This design resolves the common issues of codebook collapse and bitrate inflation, while preserving high-fidelity universal audio. Based on DistilCodec, UniTTS integrates Qwen2.5-7B Yang et al. (2024) to directly model audio token sequences, yielding natural and emotionally expressive speech synthesis. The architecture of UniTTS is illustrated in Figure 1. Our main contributions are summarized as follows:

- **DistilCodec**: We introduce a novel distillation methodology that transforms a multi-codebook NAC into a single large-vocabulary codebook. DistilCodec achieves nearly 100% code utilization and balanced distribution, enabling low-bitrate yet high-fidelity audio representation.

- **UniTTS**: We develop UniTTS, a TTS system that leverages DistilCodec for discretization and Qwen2.5-7B for audio sequence modeling. UniTTS delivers end-to-end speech synthesis with improved naturalness and expressiveness, particularly in capturing prosodic nuances.

- **Novel Audio Language Model Paradigm**: We propose a dual-phase framework for Audio Language Model (ALM): (i) *Audio Perceptual Modeling* with DistilCodec, focusing on acoustic discretization, and (ii) *Audio Cognitive Modeling* with UniTTS, aligning text and audio through pretraining, fine-tuning, and integration within the LLM. This paradigm establishes a unified path from perceptual coding to generative modeling.

## 2 RELATED WORK

### 2.1 NAC (NEURAL AUDIO CODEC)

Recently, a major research focus in NAC lies in effectively integrating prior information, such as semantic features. SpeechTokenizer Zhang et al. (2023) pioneered this direction by distilling semantic

information from HuBERT Hsu et al. (2021) representations into the first layer of its RVQ Zeghidour et al. (2021) module. This paradigm of semantic-acoustic decoupling was subsequently adopted and refined by a series of works, including including Mimi Défossez et al. (2024), X-codec Ye et al. (2025a), X-codec2 Ye et al. (2025b), and BiCodec Wang et al. (2025). Another critical research direction in NACs aims to achieve low bitrates while preserving high reconstruction quality. Representative works such as Single-Codec Li et al. (2024a), StableCodec Parker et al. (2024), BiCodec Wang et al. (2025), X-codec / X-codec2 Xin et al. (2024); Ye et al. (2025b), and WavTokenizerJi et al. (2024) have made significant strides in this area. While WavTokenizer and our DistilCodec both employ low-bitrate single-codebook NAC with full audio modeling, WavTokenizer uses a small 4k codebook. Prior research Ma et al. (2025); Zhu et al. (2024a) has established a strong correlation between codebook size and NAC performance, with models like StableCodec (15,625 codes) and X-codec2 (65,536 codes) demonstrating the substantial performance gains afforded by larger codebooks in speech discretization tasks.

## 2.2 LLM-BASED TTS

The advent of LLM-based TTS was marked by VALL-E Wang et al. (2023), which utilized Encodec Défossez et al. (2022) as its audio tokenizer and employed a multi-stage decoding strategy combining autoregressive (AR) and non-autoregressive (NAR) generation. However, this approach suffered from two key drawbacks: the LLM processed incomplete audio information, and the decoding process was overly complex. In contrast, MELL-E Meng et al. (2024) and KALL-E Zhu et al. (2024b) directly employ continuous acoustic features as input, demonstrating the viability of non-semantically aligned TTS frameworks. Nonetheless, these methods face scalability limitations in large-scale training scenarios.

To address the high computational overhead of multi-stage decoding, recent work has shifted toward single-stage paradigms. For example, Llasa Ye et al. (2025b) exploits the unified semantic-acoustic codebook of X-codec2, empirically validating the scaling laws for both data and model size in TTS tasks. Similarly, Spark-TTS Wang et al. (2025) integrates speaker characteristics and speech tokens for synthesis, while CosyVoice1.0 Du et al. (2024a) and CosyVoice2.0 Du et al. (2024b) decompose speech into speaker embeddings and semantic tokens, subsequently generating waveforms via flow matching.

## 3 METHODS

### 3.1 OVERVIEW

UniTTS distinguishes itself from existing LLM-based TTS systems by introducing DistilCodec, an audio tokenizer capable of holistically modeling universal audio. This design eliminates information loss inherent in semantic alignment while substantially reducing decoding complexity through a streamlined single-stage architecture.

The overall architecture of UniTTS consists of three key components: (1) The Encoder and Quantizer of DistilCodec serving as the Audio Tokenizer; (2) The Qwen2.5-7B Yang et al. (2024) large language model, which models the sequence of audio and text tokens; (3) The decoder from the DistilCodec model, which reconstructs the waveform from the generated tokens. Our training paradigm is organized into two distinct stages, which we term Audio Perceptual Modeling and Audio Cognitive Modeling(detailed training schema illustrated in Appendix C):

- **Audio Perceptual Modeling**: We develop DistilCodec using a novel distillation methodology called DMS (Distilling Multi-Codebook NAC to Single-Codebook NAC). DMS enables a "student" NAC to inherit the encoder and decoder parameters from a pre-trained "teacher" NAC, facilitating the training of a large single-codebook model. We train DistilCodec on a diverse universal audio dataset, resulting in a single codebook with 32,768 codes and nearly 100% utilization. Furthermore, we align the codebook's embedding dimension with that of Qwen2.5-7B (3,584), allowing us to initialize the audio embedding layer of UniTTS directly from DistilCodec's codebook, thereby passing the complete audio features to the LLM model.

- **Audio Cognitive Modeling**: The vocabulary of UniTTS is constructed by concatenating the DistilCodec codebook with the word embeddings of Qwen2.5, resulting in a vocabulary size of approximately 180,000 for UniTTS. This stage mirrors the training pipeline of modern LLMs and is divided into three phases: Pre-training, Supervised Fine-Tuning (SFT), and Alignment. Leveraging DistilCodec's ability to process complete audio information, we introduce an audio autoregressive task during pre-training, in addition to standard text-based tasks, to enhance the model's acoustic modeling capabilities. In the SFT phase, we experiment with various audio-text interleaved prompt formats to optimize performance. Finally, during the Alignment phase, we employ Direct Preference Optimization (DPO) to further refine the quality and stability of the generated speech.

## 3.2 AUDIO PERCEPTION MODELING: DISTILCODEC

The network architecture of DistilCodec follows a standard Encoder-VQ-Decoder framework similar to that proposed in Soundstream Zeghidour et al. (2021). The encoder utilizes a ConvNeXt-V2 Woo et al. (2023) structure, the vector quantization module implements a GRFVQ scheme Yang et al. (2023); Yu et al. (2021); Zeghidour et al. (2021), and the decoder is based on a ConvTranspose architecture similar to HiFiGAN Kong et al. (2020). Detailed network specifications and layer configurations are provided in Appendix B.1. The model is trained adversarially using three types of discriminators: a Multi-Period Discriminator (MPD), a Multi-Scale Discriminator (MSD), and a Multi-STFT Discriminator (MSFTFD). The detailed parameter configurations for these discriminators can be found in Appendix B.2, and the comprehensive training diagram of DisitilCodec is provided in Appendix B.3. The total loss function is a weighted sum of a Mel-spectrogram reconstruction loss, an adversarial loss, and a feature matching loss:

$$L_{\text{total}} = \lambda_{\text{mel}} L_{\text{mel}}(G) + L_{\text{adv}}(G, D) + \lambda_{\text{fm}} L_{\text{FM}}(G, D) \tag{1}$$

The Mel-spectrogram loss function is denoted as $\lambda_{\text{mel}} L_{\text{mel}}(G)$, the loss functions for the three discriminators are represented by $L_{\text{adv}}(G, D)$, while their corresponding feature map loss functions are designated as $\lambda_{\text{fm}} L_{\text{FM}}(G, D)$. The operational workflow of the DistilCodec-LSGAN-Training (DLF) is presented in AppendixB.3. The training process of DistilCodec consists of two distinct phases: teacher-training and student-training. We abbreviate the NAC in each phase as:

$$Codec_{sx} = M(N_r, N_g, N_c, N_{dim}, E_{param}^{from}, G_{param}^{from}, VQ_{param}^{from}) \tag{2}$$

$N_r$ denotes the number of NAC residual layers, $N_g$ represents the quantity of NAC Groups, and $N_{\text{dim}}$ indicates the dimension of the NAC Codebook. The parameters $E_{param}^{from}$ are initialized from the encoder of a specific codec, $G_{param}^{from}$ initialized from the Generator/Decoder of a codec, and $VQ_{param}^{from}$ correspond to the Vector Quantization (VQ) parameters from a codec. The detailed NAC architecture settings used in our training are presented in Table 1. Under the framework of the DMS (**D**istilling **M**ulti-Codebook NAC to **S**ingle-Codebook NAC), we first trained the **Teacher**$_{\text{Codec}}$ using DLF, then trained the **Student**$_{\text{Codec}}$ (namely DistilCodec) through parameter inheritance from both Encoder and Decoder of **Teacher**$_{\text{Codec}}$. The pseudo-code of DMS is presented in Algorithm 3.2.

Table 1: Settings of two stage NAC

| Codec | N-Residual | N-Group | N-Codes/Codebook | Dimension |
|-------|-----------|---------|------------------|-----------|
| Teacher-Codec | 8 | 4 | 1024 | 512 |
| Student-Codec | 1 | 1 | 32768 | 3584 |

## 3.3 AUDIO COGNITIVE MODELING

### 3.3.1 PRETRAIN

The pre-training objective is to model the joint probability distribution of audio (A) and text (T), $P(A, T)$. Using Bayes' theorem, this joint distribution decomposes as:

$$p(A \cdot T) = p(A|T) \cdot p(T) = p(T|A) \cdot p(A) \tag{3}$$

Using Bayes' theorem, this objective can be decomposed into three distinct training tasks:

---

**Algorithm 1** DMS: Distilling Multi-Codebook NAC to Single-Codebook NAC via parameter inheritance)

---

1: **Step 1:** Initializing Teacher$_{\text{codec}}$:
   Teacher$_{\text{codec}}$ = Codec$_{s1}(8, 4, 1024, 512, E_{\text{param}}^{\text{scratch}}, G_{\text{param}}^{\text{scratch}}, VQ_{\text{param}}^{\text{scratch}})$
2: **Step 2:** Teacher$_{\text{codec}}$ training with DLF
3: **Step 3:** Initializing Student$_{\text{codec}}$:
   Student$_{\text{codec}}$ = Codec$_{s2}(1, 1, 35768, 3584, E_{\text{param}}^{\text{teacher}}, G_{\text{param}}^{\text{teacher}}, VQ_{\text{param}}^{\text{scratch}})$
4: **Step 4:** Student$_{\text{codec}}$ training with DLF
5: **Output:** DistilCodec = Student$_{\text{codec}}$

---

- $p(A)$: The model learns to predict the next audio token given the preceding audio tokens.
- $p(T)$: This is the standard causal language modeling task, which preserves text capabilities.
- $p(A|T)\&p(T|A)$: The model learns to generate audio conditioned on text and text conditioned on audio, thereby learning the alignment between the two modalities.

Consequently, our audio modality pre-training extends the text modality by incorporating audio modality auto-regression and text audio alignment tasks. Additionally, as demonstrated in Appendix C.10, our experiments reveal that audio modeling presents greater spatial complexity and implementation challenges compared to text modeling. The scarcity of high-quality text-audio paired data further necessitates the integration of a universal audio autoregressive task, which proves beneficial for enhancing final model performance. Within this pre-training phase, we designed a multi-stage training approach.

- Stage 1: Training on text, universal audio, and limited paired text-audio data to establish audio modeling relationships. However, introducing audio data to a text-pretrained model induced modality competition, degrading text generation. This outcome directly motivated the subsequent training in Stage 2.
- Stage 2: Augmenting with text-based instruction datasets alongside universal audio and text-audio pairs to restore text capabilities, further enhancing the model's text generation capabilities (details in Appendix C.9). We also extended the context window from 8,192 to 16,384 tokens to support longer sequences.

### 3.3.2 SFT

The quality of instruction tuning data significantly impacts the final model performance. However, Open-source text-speech datasets suffer from: (1) noisy ASR-derived text and (2) prolonged silences in podcast/audiobook excerpts, which are unsuitable for TTS. To address these issues, we developed a data filtering algorithm based on a composite quality score. This score is derived from metrics assessing both text accuracy and audio quality. Text accuracy is evaluated by generating reference text using the Paraformer Gao et al. (2022) and Whisper Radford et al. (2023) models and computing the Character Error Rate (CER). Audio quality is assessed using the DNSMOS P.835 OVRL metric Reddy et al. (2022). For each text-audio pair, this score is calculated as:

$$\text{quality}(x_i) = \text{dnsmos}(x_i) - \text{cer}(x_i) \tag{4}$$

Where $x_i$ denotes the index of the sample, $quality(x_i)$ represents the DNSMOS score for sample $x_i$, and $cer(x_i)$ represents the CER score for sample $x_i$. Samples are subsequently ranked in descending order based on their quality scores, and a predetermined number of the highest-ranked samples are selected for inclusion in the training set. The pseudocode is presented in Appendix C.2.

Our experiments demonstrated that incorporating text instruction data and the long-cot instruction dataset not only enhanced the model's text understanding capability but also led to further improvements in audio generation quality. The templates for text-to-speech (TTS) conversion, text dialogue, and long-cot instructions are provided in Appendix C.1.

### 3.3.3 ALIGNMENT

Following SFT, the model occasionally exhibited undesirable artifacts such as prosodic lengthening and repetition, which we hypothesize stems from the relative scarcity of speech data compared to text

data during pre-training. As evidenced by the pre-training loss curve in Fig. 7, the audio generation loss persists at elevated levels while maintaining a consistent downward trajectory.

To mitigate these issues and further enhance generation stability, we employed a preference optimization algorithm. While Direct Preference Optimization (DPO) is a common choice, its vanilla implementation can be susceptible to mode collapse in ultra-long sequence tasks like TTS. Consequently, we opted for Linear Preference Optimization (LPO), a more stable alternative, whose training objective is defined as:

$$L_{\text{lpo}} = \gamma \cdot (x_1^{\text{ste}} + x_2^{\text{ste}}) + \lambda \max(0, -\log x_1^{\text{ste}}) \tag{5}$$

In formula 5, $x_1^{\text{ste}}$, $x_2^{\text{ste}}$, $\gamma$, $x_1$, $x_2$ are detailed in Appendix C.5.

## 4 EXPERIMENTS

This section evaluates the empirical performance of our proposed methods. We first assess the capabilities of the DistilCodec audio tokenizer and subsequently analyze the end-to-end performance of the UniTTS system.

### 4.1 EXPERIMENTAL SETUP FOR DISTILCODEC

The training corpus for DistilCodec comprised a diverse, 100,000-hour universal audio dataset, including Chinese and English audiobooks, general speech, music, and sound effects. Detailed data distributions are provided in Table 15.

The training was conducted on a cluster of 5×8 A100 GPUs. The two-stage DMS training process involved training the Teacher Codec for 5 epochs and the Student Codec (DistilCodec) for 3 epochs. Both models were optimized using the AdamW optimizer with hyperparameters specified in Appendix B.3.

### 4.2 DISTILCODEC EVALUATION

We compared the codebook utilization of our method with WavTokenizer Ji et al. (2024), particularly in scenarios with large codebooks. We employed the LibriSpeech-Clean dataset to evaluate the codebook utilization rate of DistilCodec, with the corresponding experimental results presented in Table 2. From Table 2, it can be observed that while wavtokenizer achieves only a 68% codebook utilization rate with a codebook size of 8192, this rate drops to just 27% when the codebook size increases to 16384. In contrast, DistilCodec effectively solves the codebook collapse problem by achieving near-optimal codebook utilization (approaching 100%) with a codebook size of 32768. Additionally, we conducted a comprehensive comparative analysis of DistilCodec's speech reconstruction capabilities using the LibriSpeech-Clean-Test benchmark, and the results are shown in Table 3.

Table 2: Comparison of codebook utilization across different models

| Model | Codebooks | Codebook Usage(%)↑ |
|---|---|---|
| WavTokenizer | 8192 | 68 |
| WavTokenizer | 16384 | 27 |
| DistilCodec | 32768 | **98.2** |

Since DistilCodec was trained on universal audio, we first employed UTMOS Saeki et al. (2022) for automatic quality assessment. However, the universal audio test set(a self-constructed Universal Audio dataset) received an unreliable low score (1.89), indicating UTMOS's inadequacy for universal audio evaluation. We therefore conducted a Mean Opinion Score (MOS) evaluation, which consists of:

**Evaluation Dataset**: We selected 98 universal audio clips, comprising Chinese and English audiobooks, streaming media audio content, and sound effects.

**Evaluation Protocol**:

Table 3: Comparison of various models based on codebook size, token rate, bandwidth, and quality metrics

| Model | Codebook Size | Nq | Token Rate (TPS) | Bandwidth (bps) | STOI ↑ | PESQ ↑ | UTMOS ↑ |
|---|---|---|---|---|---|---|---|
| Encodec | 1024 | 8 | 600 | 6000 | 0.94 | 2.75 | 3.07 |
| DAC | 1024 | 12 | 600 | 6000 | 0.95 | 4.01 | 4.00 |
| Encodec | 1024 | 2 | 150 | 1500 | 0.84 | 1.56 | 1.58 |
| Mimi | 2048 | 8 | 100 | 1100 | 0.91 | 2.25 | 3.56 |
| BigCodec | 8192 | 1 | 80 | 1040 | 0.94 | 2.68 | 4.11 |
| DAC | 1024 | 2 | 100 | 1000 | 0.73 | 1.14 | 1.29 |
| SpeechTokenizer | 1024 | 2 | 100 | 1000 | 0.77 | 1.25 | 2.28 |
| X-codec | 1024 | 2 | 100 | 1000 | 0.86 | 2.33 | 4.21 |
| WavTokenizer | 4096 | 1 | 75 | 900 | 0.89 | 2.14 | 3.94 |
| X-codec2 | 65536 | 1 | 50 | 800 | 0.92 | 2.43 | 4.13 |
| StableCodec | 15625 | 2 | 50 | 697 | 0.91 | 2.24 | 4.23 |
| Single-Codec | 8192 | 1 | 23.4 | 304 | 0.86 | 1.88 | 3.72 |
| BiCodec | 8192 | 1 | 50 | 650 | 0.92 | 2.51 | 4.18 |
| **DistilCodec** | 32768 | 1 | 93 | 1300 | 0.93 | 2.02 | 3.75 |

- Speech Clarity: Subjective rating (0-5 scale) of vocal articulation quality.
- Background Clarity: Subjective rating (0-5 scale) of environmental sound.

Table 4 presents the MOS results for universal audio under this evaluation framework. Evaluation results demonstrate that DistilCodec achieves superior scores in both speech clarity and background clarity, indicating its capability for universal audio reconstruction.

Table 4: Comparison of MOS scores

| Assessment Items | Synthetic Speech | GT |
|---|---|---|
| Speech Clarity | 4.689 | 4.945 |
| Background Audio Clarity | 4.768 | 4.927 |
| Average Score | 4.728 | 4.936 |

## 4.3 TTS EXPERIMENTS

### 4.3.1 EXPERIMENTAL DETAILS

**Pretraining**: The pre-training corpus consisted of 322B tokens, comprising a mix of universal audio data, text data, and aligned text-audio pairs. The dataset sources included Libriheavy Kang et al. (2024), WenetSpeech4TTS Ma et al. (2024), Emilia He et al. (2024), and our proprietary collections. The text data consists of our self-collected datasets, Infinity-Instruct, and SkyPile-150B. The complete data distribution is detailed in Appendix C.3.

The pre-training was conducted in two stages: the first stage employs a cosine-annealed learning rate schedule decaying from 1e-4 to 2e-5 with 10% warmup proportion, employing 8,192 windows and a batch size of 256 for training efficiency; the second stage continues with a finer learning rate decay from 2e-5 to 9e-6 while extending the window length to 16,384 and maintaining the same batch size.

**Supervised Fine-Tuning**: For the Supervised Fine-Tuning (SFT) phase, we employed a multi-task learning framework to simultaneously enhance the model's conversational understanding and audio generation capabilities. The training was conducted on a composite dataset comprising text instructions, long-cot reasoning data, and curated TTS pairs. Notably, the total audio data for this phase amounted to approximately **948** hours (significantly lower than LLaSA's 250K hours and Spark-TTS's 100K hours) as detailed in Appendix C.4.

During the SFT phase, we adopt a cosine learning rate schedule that decays from 9e-6 to 5e-6, with a context window length of 8,192 and a batch size of 128.

**Alignment**: To further stabilize the model's performance, we employed LPO training. The distribution of the original alignment data used for LPO is provided in Appendix C.5:

Next, following the preference pair generation method described in LPO, we used each sample's prompt as input to generate three candidate responses. These candidates were then paired with the sample's reference answer to form three preference pairs, which constitute the training data for LPO. The training parameters for LPO are listed in Table 19.

### 4.3.2 TEXT-TO-SPEECH EVALUATION

We conducted a comprehensive evaluation of the UniTTS to assess its audio generation performance. Unlike existing approaches that separately model acoustic and semantic features, UniTTS utilizes DistilCodec to holistically process the audio signal. The evaluation was performed on models at two key stages of our three-phase training pipeline: the model after Supervised Fine-Tuning (UniTTS-SFT) and the model after the alignment stage (UniTTS-LPO).

For a rigorous evaluation, we compared UniTTS against state-of-the-art methods, including:CosyVoice2, Spark-TTS, LLaSA, F5-TTS Liao et al. (2024), Fish-Speech Chen et al. (2024), IndexTTS Deng et al. (2025).

We constructed a diverse evaluation dataset inspired by the methodology of Llasa Ye et al. (2025b). This benchmark includes challenging scenarios such as emotions, rare characters, tongue twisters, interjections, audiobooks, and conversational quirks like stuttering. Furthermore, to assess timbre replication across different demographics, the dataset features curated voice samples from four distinct age-gender categories: children, adult males, adult females, and elderly speakers. The detailed evaluation protocol is described in Appendix C.6, with experimental results presented in Table 5.

Table 5: Comparison of Mean Opinion Score between different TTS models.

| Model | Fidelity | Stability | Naturalness | Emotional expressiveness | Average Score |
|---|---|---|---|---|---|
| Cosyvoice2 | 4.80 | 5 | 4.89 | 4.11 | 4.70 |
| SparkTTS | 4.89 | 5 | 4.89 | 4.26 | 4.76 |
| Llasa | 4.74 | 4.91 | 4.91 | 4.11 | 4.67 |
| F5-TTS | 4.94 | 5 | 4.89 | 3.97 | 4.70 |
| Fish Speech | 4.89 | 5 | 4.83 | 4.29 | 4.75 |
| IndexTTS | 4.69 | 4.83 | 4.89 | 4.31 | 4.68 |
| UniTTS-SFT | 4.43 | 5 | 4.77 | 4.23 | 4.61 |
| UniTTS-LPO | 4.80 | 4.97 | 4.94 | 4.60 | **4.83** |

The results presented in Table 5 demonstrate that UniTTS-LPO achieves comprehensive improvements over UniTTS-SFT in emotional expressiveness, fidelity, and naturalness, thereby validating the effectiveness of the LPO training methodology. UniTTS-LPO's performance superiority stems from its DistilCodec-powered holistic modeling of prosodic-timbral-emotional features and diverse unsupervised training, enabling state-of-the-art emotion-aware speech synthesis.

### 4.3.3 ABLATION ANALYSIS

To investigate the factors influencing the performance of UniTTS, we conducted comprehensive ablation experiments. We employed the CER (Character Error Rate) metric from seed-tts-eval as our evaluation criterion for the ablation study.

Our Baseline configuration is as follows: A mixture of a 1.2M TTS dataset, a 181K text dataset, and a 55K long-long-cot dataset. The PROMPT1 structure is used, and details for PROMPT1, PROMPT2, and PROMPT3 can be found in Appendix C.7.

Building on this baseline, we conducted a series of ablation studies to investigate the factors that influence the model's performance.

Table 6: Comparison of CER for different ablation tests

| Model | CER ↓ |
|---|---|
| baseline | **3.466%** |
| w/o reference text in prompt | 5.5045% |
| + ASR instruction data | 3.582% |
| w/o text instruction data | 3.7025% |
| reverse reference text in prompt | 3.7395% |
| w/o sft data filter algorithm | 5.15% |

**Impact of Text Instructions on Model Performance:** We removed the text-instruction and long-cot dataset, denoted as *w/o text instruction data*. According to Table 6, the experimental results showed that incorporating text instructions improves audio generation quality.

**Influence of Instruction Templates on Model Performance:** We removed the text corresponding to the reference audio(denoted as *w/o reference text in prompt*). The prompt used was PROMPT2, with further details provided in Appendix C.7. As shown in Table 6, including both the example audio and its associated text in the prompt template yields a performance improvement of approximately 2.1-point. We character order of the reference audio's text is reversed within the prompt, PROMPT2 is used. We swapped the order of the reference audio and the winning text, denoted as *reverse the order reference audio and text*. The prompt used was PROMPT3, Compared to the baseline, this change still harms the final result.

**Compatibility Between TTS and ASR Tasks:** We added an ASR task to the training set to check for task compatibility, which is denoted as + *ASR Instruction Data*. According to Table 6, we found that including the ASR task degraded the performance of the TTS generation. This is likely due to the inherent competition between the ASR and TTS objectives, exacerbated by insufficient model training and limited model capacity.

**Influence of data of quality:** We removed the SFT data filtering algorithm(Appendix C.2), which we denote as *w/o sft data filter algorithm*. According to Table 6, the results show that this change led to a 1.68-point decrease in model performance, demonstrating the critical importance of data quality.

## 5 CONCLUSION

We present DistilCodec, an innovative audio codec distillation framework that effectively compresses multi-codebook architectures (RVQ/GVQ) into a single, high-capacity codebook (32,768 codes) while achieving 100% code utilization and mitigating mode collapse. Leveraging Distil-Codec as our audio encoder, we develop UniTTS – a Qwen2.5-7B-based model trained through a tri-task pretraining regimen encompassing speech autoregression, text autoregression, and cross-modal speech-text generation, empirically demonstrating reduced dependence on precisely aligned text-speech data. Our experiments reveal UniTTS's capability to generate semantically coherent, emotionally expressive speech, with quantitative evidence showing that text instruction data enhances audio quality. Preliminary observations of text-audio modality competition suggest future exploration of MoE architectures and inter-modal optimization strategies.

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

## A    LARGE LANGUAGE MODEL USE DECLARATION

In preparing this work, the authors used a large language model (LLM), specifically DeepSeek, to enhance the clarity and fluency of certain sections. The LLM was used for grammar checking, sentence restructuring, and improving academic phrasing in the Introduction, Methods, and Experiments sections. The authors have thoroughly reviewed and edited the output to ensure the integrity and accuracy of the content and take full responsibility for the entire publication.

## B    DISTILCODEC

### B.1    MODEL STRUCTURE OF DISTILCODEC

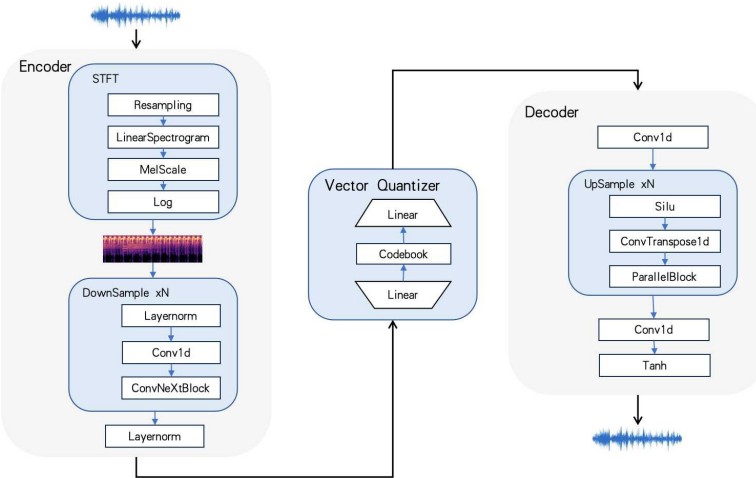

Figure 2: The detailed network architecture of DistilCodec.

The detailed network architecture of DistilCodec is illustrated in Figure 2. DistilCodec consists of three key components:

**Audio Encoder**: The Mel-spectrogram parameters are specified in Table 7, while the encoder configurations are detailed in Table 8.

**Vector Quantizer**: The quantizer and factor settings are provided in Table 9.

**Audio Decoder**: The specific parameters are listed in Table 10.

Table 7: Mel-spectrogram STFT settings.

| Configuration Item | Value |
|---|---|
| Sampling Rate | 24000 |
| Segment Size | 72000 |
| Mel Channels | 128 |
| Hop Size | 256 |
| Window Size | 1024 |
| FMin | 0 |
| FMax | 12000 |

Table 8: Encoder settings of DistilCodec.

| Configuration Item | Value |
|---|---|
| Input Channels | 128 |
| Number of Downsampler | 4 |
| Downsampler Depth | [3, 3, 9, 3] |
| Downsampler Output Dims | [256, 512, 768, 1024] |
| ConvNeXt Drop Rate | 0.2 |
| Conv Kernel Size | 7 |

Table 9: Settings for DistilCodec's Vector Quantizer.

| Configuration Item | Value |
|---|---|
| Number of Residual | 1 |
| Number of Group | 1 |
| Number of Codebook | 1 |
| Dimension of Code | 3584 |
| Number of Codes | 32768 |
| EMA Decay Rate | 0.8 |
| Conv Kernel Size | 7 |
| Pre Factorized Dense | [1024, 3584] |
| Post Factorized Dense | [3584, 1024] |

## B.2 DISCRIMINATORS OF DISTILCODEC

During the training phase, the GAN-based framework employs three discriminators:

**Multi-period discriminator**: Detailed parameters are provided in Table 11.

**Multi-scale discriminator**: Detailed parameters are listed in Table 12.

**Multi-STFT discriminator**: Specific configurations can be found in Table 13.

## B.3 DISTILCODEC TRAINING FRAMEWORK

Fig.3 illustrates the comprehensive training diagram of DisitilCodec. The configuration of optimizer parameters during the training of DistilCodec can be found in Table 14, while the LSGAN training pseudocode for DistilCodec (DLT) is outlined in Algorithm 2. The training data of DistilCodec is illustrated in Table 15.

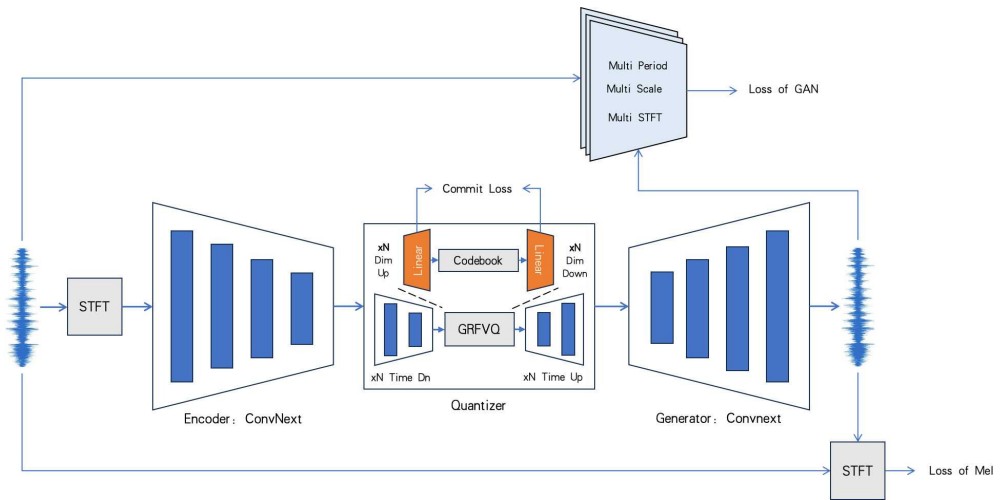

Figure 3: Training Diagram of DisitilCodec.

---

**Algorithm 2** DistilCodec's LSGAN Training Pseudo Code

---

**Require:** $Audio_{universal}$
    **for** epoch $= 0, 1, 2, \ldots$ **do**
        **for** step $= 0, 1, 2, \ldots$ **do**
            $e_f = \text{encoder}(x)$
            $(\text{quantized\_f}, l_{commit}) = \text{quantizer}(e_f)$
            $y_g = \text{generator}(\text{quantized\_f})$
            $l_{mel} = \text{multi\_scale\_mel\_loss}(y, y_g.\text{detach}())$
            $l_{gan} = \text{gan\_loss}(y, y_g)$
            $l_{total} = l_{mel} + l_{commitment} + l_{gan}$
            $\text{BACKWARD}(l_{total})$
        **end for**
    **end for**
**Ensure:** Trained Audio Codec

---

Table 10: Decoder settings of DistilCodec.

| Configuration Item | Value |
|---|---|
| Number of Upsampler | 5 |
| Upsampler Rates | [8, 4, 2, 2, 2] |
| Resblock Kernel Sizes | [3, 7, 11] |
| Resblock Dilation Sizes | [[1, 3, 5], [1, 3, 5], [1, 3, 5]] |
| ConvNeXt Drop Rate | 0.2 |
| Conv Kernel Size | 7 |
| Pre Conv1d Kernel Size | 13 |
| Post Conv1d Kernel Size | 13 |

Table 11: Multi Period Discriminator parameter settings of DistilCodec.

| Configuration Item | Value |
|---|---|
| Number of Period Discriminator | 5 |
| Periods | [5, 8, 13, 19, 30] |
| Kernel Size | 5 |
| Stride | 3 |

## C    UNITTS

Fig.4 illustrates the training schema of UniTTS and DistilCodec.

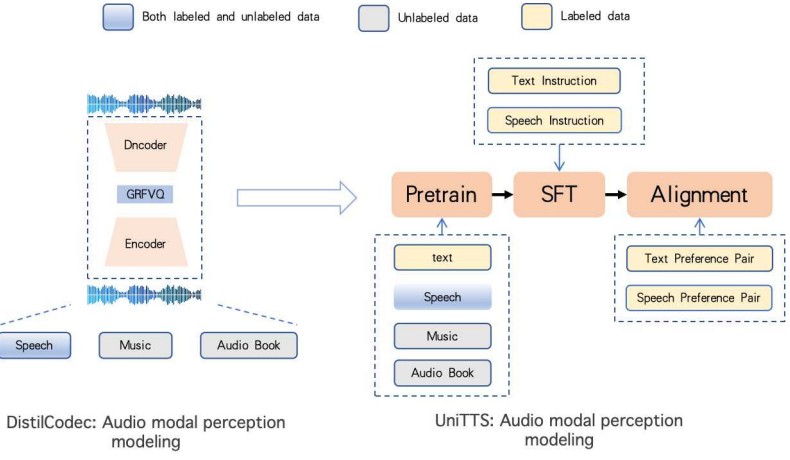

Figure 4: Training schema of UniTTS and DistilCodec. DistilCodec consists of three core components: an Encoder, GRFVQ, and a Decoder, trained on universal audio data. The training process of UniTTS follows a methodology analogous to that of Large Language Models (LLMs), comprising three stages: Pretraining, Supervised Fine-Tuning (SFT), and Alignment. Notably, the pretraining phase utilizes universal audio as part of its training data.

### C.1    UNITTS PROMPT

The TTS prompt template, long-CoT prompt template, and text dialogue template are detailed in Fig.5

### C.2    UNITTS DATA FILTERING ALGORITHM

The sft data filtering algorithm of UniTTS is illustrated in Algorithm 3.

Figure 5: Inference Prompt Template

---

**Algorithm 3** The data filtering algorithm based on quality scores

---

**Require:** Text-Audio Alignment Dataset $D(x, y)$ with $N$ samples
**Ensure:** scores
1: Initialize empty list *scores* = []
2: **for** $i \leftarrow 1$ to $N$ **do**
3:     $x_i, y_i = D(i)$
4:     $dnsmos(x_i) = DNSMOS\ P.835\ OVRL(x_i, y_i)$
5:     $transcribed\_text1 = paraformer(y_i)$
6:     $transcribed\_text2 = whisper(y_i)$
7:     $cer(x_i) = cal\_cer(transcribed\_text1, transcribed\_text2)$
8:     $quality(x_i) = dnsmos(x_i) - cer(x_i)$
9:     $vad\_proportion = cal\_vad(y_i)$
10:     **if** $vad\_proportion > 0.14$ **then**
11:         **continue**
12:     **end if**
13:     scores.append$((x_i, y_i, quality(x_i)))$
14: **end for**
15: Sort scores by $q$ in descending order ▷ In-place sort (high to low)
16: **return** scores

---

Table 12: Multi Scale Discriminator parameter settings of DistilCodec.

| Configuration Item | Value |
|---|---|
| Number of Period Discriminator | 5 |
| Periods | [5, 8, 13, 19, 30] |
| Kernel Size | 5 |
| Stride | 3 |

Table 13: Multi STFT Discriminator parameter settings of DistilCodec.

| Configuration Item | Value |
|---|---|
| Number of STFT Discriminator | 5 |
| Number of FFTs | [1024, 2048, 512, 256, 128] |
| Hop Lengths | [256, 512, 128, 64, 32] |
| Window Lengths | [1024, 2048, 512, 256, 128] |
| Number of Filter | 32 |

### C.3 DISTRIBUTION OF PRETRAINING DATA FOR UNITTS

Distribution of pretraining data is illustrated in Table 16

### C.4 DISTRIBUTION OF SFT DATA

Distribution of sft data is illustrated in Table 18

### C.5 LINEAR PREFERENCE OPTIMIZATION

In formula 5, $x_1^{\text{ste}}$, $x_2^{\text{ste}}$, $\gamma$, $x_1$, $x_2$ are shown in the following:

$$x_1^{\text{ste}} = r_1 \cdot \max(0, x_1 - x_2.\text{detach}() - \frac{1}{2\beta}) \tag{6}$$

$$x_2^{\text{ste}} = r_2 \cdot \max\left(0, x_{1.\text{detach}()} - x_2 - \frac{1}{2\beta}\right) \tag{7}$$

$$\gamma = 2\beta \cdot \frac{2}{r_1 + r_2} \tag{8}$$

$$x_1 = \frac{\Pi_\theta(y_w|x)}{\Pi_{\text{ref}}(y_w|x)} \tag{9}$$

$$x_2 = \frac{\Pi_\theta(y_l|x)}{\Pi_{\text{ref}}(y_l|x)} \tag{10}$$

Here, $\Pi_\theta$ denotes the policy to be optimized, and $\Pi_{\text{ref}}$ represents the reference model, which is equivalent to the UniTTS model after Supervised Fine-Tuning (SFT). The variables $y_w$ and $y_l$ correspond to a pair of samples where, given the input prompt $x$, $y_w$ demonstrates superior performance compared to $y_l$. LPO's hyperparameter is illustrated in Table 19.

Table 14: AdamW Experiment Parameter Settings

| Parameter | Value |
|---|---|
| $\beta_1$ | 0.5 |
| $\beta_2$ | 0.9 |
| LR Decay | 0.98 |
| Weight Decay | 0.001 |

Table 15: Distribution of DistilCodec training data

| Data Category | Data Size (in hours) |
|---|---|
| Chinese Audiobook | 38000 |
| Chinese Common Audio | 20000 |
| English Audiobook | 10000 |
| English Speech | 30000 |
| Music | 2000 |
| Total | 100000 |

Table 16: Distribution of pretraining data

| Data Type | Data Size (B) |
|---|---|
| Text Data | 140 |
| Text-Audio Alignment Data | 82 |
| Audio Data | 100 |
| Total | 322 |

## C.6 EVALUATION CRITERIA FOR UNITTS

To assess model performance, we conducted a Mean Opinion Score (MOS) evaluation on the test set constructed in Section 3.4 using the following criteria (rated on a 0-5 scale).

1) Fidelity: The audio accurately reproduces the original sound characteristics, including timbre and pitch alignment with ground truth recordings.

2) Stability: The audio playback exhibits no artifacts such as stuttering, frame skipping, or abrupt termination.

3) Naturalness: The output demonstrates human-like speech/instrument production without robotic artifacts or unnatural prosody.

4) Emotional expressiveness: The audio effectively conveys intended emotional states (e.g., joy, sadness, anger) with appropriate vocal/instrumental cues.

## C.7 CONFIGURATION OF TTS ABLATION EXPERIMENTS

We introduce three distinct prompt configurations: PROMPT1 incorporates both the reference audio exemplar and its corresponding transcript, with the transcript preceding the audio; PROMPT2 inverts this sequence by presenting the audio prior to the transcript; PROMPT3 excludes the transcript while retaining the audio exemplar from PROMPT1. The prompt configurations are illustrated in Figure 6.

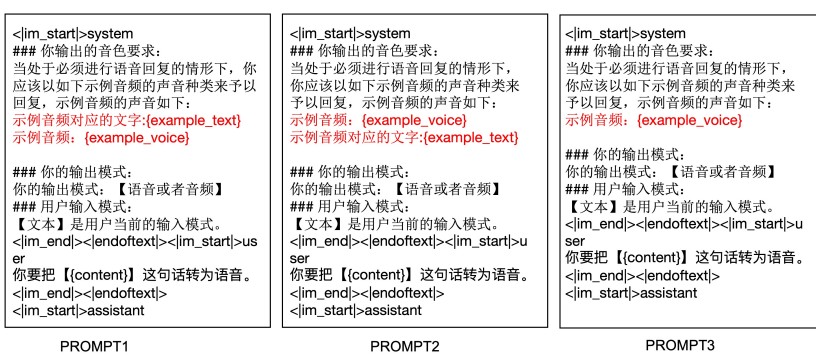

Figure 6: Prompt configurations

Table 17: Distribution of sft data

| Data Type | Number of Samples |
|---|---|
| Text Data | 181K |
| Long-cot Dataset | 55K |
| Text-Audio Alignment Data | 401K |
| Total | 637K |

Table 18: Distribution of lpo data

| Data Type | Number of Samples |
|---|---|
| General SFT Data | 100K |
| Long-cot Dataset | 45K |
| Text-Audio Alignment Data | 300K |
| Total | 445K |

### C.8 PRE-TRAINING LOSS OF STAGE 1

We present the loss curve from the first phase of pre-training. As shown in Fig.7, the model's loss remains relatively high but exhibits a clear downward trend, indicating that the computational resources and training data during the pre-training stage were still insufficient.

### C.9 TEXT CAPABILITY TESTING OF UNITTS

The pre-training process comprises two stages. In Stage 1, we observed that introducing the audio modality degraded text generation performance, leading to overall task deterioration due to limited model capacity and competition between modalities. We hypothesized that strengthening the model's textual instruction-following ability could enhance both contextual understanding and audio generation quality. This hypothesis was empirically validated in the SFT experiments (4.3.3), where improvements in audio generation were noted after bolstering text instruction capabilities.

In Stage 2 of pre-training, we employed a data augmentation strategy by incorporating text-based instruction datasets. This approach substantially restored text generation performance, as demonstrated by the comparative analysis in Table 20. However, it is worth noting that code instruction and mathematics-related datasets were excluded from this phase, which may explain the suboptimal results in human evaluation metrics and mathematical reasoning tasks.

### C.10 EXPERIMENTAL VALIDATION OF THE TEXT-TO-INSTRUCTION DATA ALIGNMENT FRAMEWORK

In our study, we investigated different training methodologies for Text-to-Speech (TTS) models. We found that a pure Supervised Fine-Tuning (SFT) approach, which we refer to as the "pure-sft" model, without any pre-training, yielded poor results. As shown in Table 21, using 6.2 million text-audio pairs for SFT resulted in a high Character Error Rate (CER) of 18.18% and subpar audio quality. Our analysis revealed that when the model was given the same text to synthesize twice, only 5% of the generated audio tokens were identical. This highlights the significant challenge of audio modeling, as the model must navigate a vast audio sequence space to capture nuances like prosody, timbre, and semantic information, a task far more complex than modeling a text sequence.

To address this challenge and the scarcity of high-quality text-audio alignment data, we adopted a pre-training approach inspired by codec systems that use unlabeled audio data. During the pre-training phase, we used 100 billion unlabeled audio samples for training. This allowed the model to first learn robust audio modeling capabilities through self-supervised learning. Following this, we performed SFT using a much smaller dataset of just 401k text-audio aligned samples. This two-stage approach resulted in a significantly improved CER of 3.43%, demonstrating that a strategic pre-training phase focused on audio modeling is crucial for achieving superior performance with a limited amount of SFT data.

Table 19: LPO Parameter values

| item | value |
|------|-------|
| $\beta$ | 0.2 |
| $\delta$ | 10.0 |
| $r_1$ | 1.0 |
| $r_2$ | 0.4 |
| Max LR | 8e-7 |
| Min LR | 5e-7 |
| Global Batchsize | 120 |

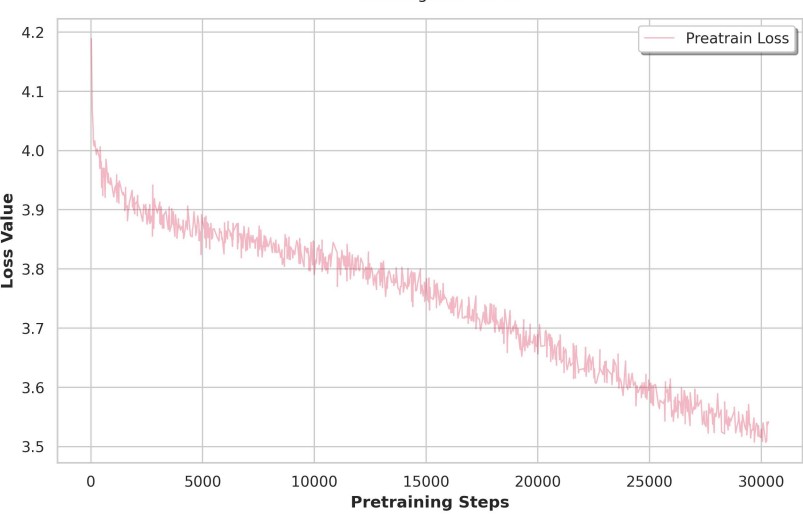

Figure 7: Stage 1 Pre-training Loss

Table 20: Performance comparison across different datasets and stages

| datasets | qwen2.5-7b-base-opencompass | pretrain-stage1 | pretrain-stage2 |
|----------|------------------------------|-----------------|-----------------|
| MMLU | 74.26 | 47.12 | 52.44 |
| ARC-C | 59.66 | 32.0 | 37.97 |
| Winogrande | 68.98 | 59.12 | 58.96 |
| Hellaswag | 86.63 | 63.24 | 62.18 |
| GPQA | 39.39 | 23.23 | 25.76 |
| MATH | 51.2 | 2.86 | 7.78 |
| GSM8K | 79.45 | 19.18 | 64.97 |
| HumanEval | 77.44 | 10.98 | 14.63 |

Table 21: Comparison of CER for different Models

| Model | CER |
|-------|-----|
| PURE_SFT | 18.18% |
| UniTTS-SFT | 3.43% |

