# OpenReview forum: "UniTTS: Towards End-to-End Speech Synthesis with Joint Acoustic-Semantic Modeling"
_ICLR.cc/2026/Conference — ICLR 2026 Conference Withdrawn Submission_

### Official Review · Reviewer_6mX9 · 2025-10-19

**Soundness:** 3
**Presentation:** 3
**Contribution:** 2
**Rating:** 6
**Confidence:** 3

**Summary:**

This paper presents UniTTS, a LLM-based TTS framework with a novel neural audio codec called DistilCodec. DistilCodec employs a single codebook aligned with Qwen2.5-7B, enabling direct initialization of the audio embedding layer. UniTTS introduces a joint audio-text vocabulary and explores audio-text interleaved prompting. Experimental evaluations on LibriSpeech-Clean demonstrate that DistilCodec achieves approximately 98.2% codebook utilization. Moreover, applying Linear Preference Optimization (LPO) further enhances the mean opinion score (MOS) of UniTTS.

**Strengths:**

This paper makes a valuable contribution to the field of text-to-speech (TTS) by introducing large-scale, LLM-based training. The proposed alignment with Qwen-2.5 is both interesting and promising.

**Weaknesses:**

The primary concerns are related to limited experimental validation and the resulting performance.

**Ablation Study** Table 6 presents ablation results. However, the analysis is constrained to a single metric (CER). Since TTS models typically involve a trade-off between accuracy and speaker identity preservation, it is essential to include additional metrics such as UTMOS, Speaker Similarity (Speaker SIM), and Word Error Rate (WER) for a comprehensive evaluation.

**Degraded Audio Quality** In Table 3, the PESQ score of the proposed method is significantly lower compared to other neural audio codec (NAC) models. This raises a critical concern regarding audio quality degradation. Further explanation or analysis is necessary to understand why this performance gap occurs.

**Questions:**

1. Why does the ablation study in Table 6 focus solely on CER? Additional metrics are needed.

2. Why does the proposed method yield significantly lower PESQ scores compared to other neural audio codec (NAC) models in Table 3?

---

### Official Review · Reviewer_s3oD · 2025-10-29

**Soundness:** 2
**Presentation:** 3
**Contribution:** 2
**Rating:** 4
**Confidence:** 4

**Summary:**

This paper addresses two primary challenges in LLM-based Text-to-Speech :the "codebook collapse" phenomenon that occurs when training single-codebook NACs with large vocabularies, and 2) the information bottleneck created by models that decouple speech into separate semantic and acoustic tokens. To solve this, the paper proposes two key contributions: 1. DistilCodec: A single-codebook NAC with a large vocabulary. It is trained using a new method called DMS (Distilling Multi-Codebook NAC to Single-Codebook NAC), where a "student" single-codebook codec inherits the encoder and decoder parameters from a pre-trained "teacher" multi-codebook codec. This method reportedly achieves near-100% codebook utilization (98.2%), effectively solving the codebook collapse problem. 2. UniTTS: A large-scale (7B) TTS model built upon DistilCodec. Unlike decoupled models, UniTTS performs joint acoustic-semantic modeling by treating speech as a single sequence of holistic audio tokens. It employs a three-stage training pipeline: Pre-training, SFT and RL. The authors demonstrate that DistilCodec achieves high codebook utilization and that the final UniTTS-LPO model achieves SOTA MOS scores in zero-shot voice cloning and emotional expressiveness.

**Strengths:**

1. The core idea of inheriting pre-trained encoder/decoder weights from a multi-codebook "teacher" to stabilize the training of a large single-codebook "student". It directly addresses the well-known problem of codebook collapse, and the reported 98.2% utilization of a 32k codebook is a strong result.
2. By using a single, unified audio token stream, UniTTS can be pre-trained on diverse, unlabeled "universal audio" (including music and sound effects), which is a significant advantage for data scaling and capturing non-linguistic information.

**Weaknesses:**

1. There appears to be a contradiction in the evaluation of DistilCodec. The objective reconstruction metrics (Table 3) are notably worse than many baselines. For example, DistilCodec scores a PESQ of 2.02 and UTMOS of 3.75, while X-codec2 achieves 2.43/4.13 and BiCodec achieves 2.51/4.18. The paper dismisses this by stating UTMOS is unreliable for universal audio, but this doesn't explain the low PESQ.
2. The paper introduces "Linear Preference Optimization (LPO)" and claims it is a "more stable alternative" to DPO, especially for long sequences. However, this claim is presented without any citation, theoretical justification, or empirical evidence (e.g., an ablation study comparing LPO to standard DPO).

**Questions:**

1. The central claim is that DMS enables a large 32k single codebook. How does UniTTS perform if you use DMS to train a smaller codebook? This would demonstrate whether the large codebook size is a critical factor for the model's high performance.
2. Why does initializing a single-layer speech codec with an encoder and decoder pre-trained via a multi-codebook GRVQ teacher solve the codebook collapse problem? Codebook collapse originates from the VQ learning process, but the student's VQ parameters are trained from scratch. While this initialization provides a better representation for the encoder/decoder, it's not clear how this guarantees the avoidance of collapse in the new VQ layer. The paper lacks a sound theoretical explanation for this.
3. Was DPO attempted, and if so, did it exhibit the instability mentioned?
4. DistilCodec's objective reconstruction metrics (Table 3) are not superior to prior single-codebook codecs (e.g., X-codec2). This makes it unconvincing that the subjective improvements in UniTTS (Table 5) stem directly from DistilCodec itself, rather than from the large-scale, three-stage training pipeline. To isolate the benefit of DistilCodec, could you conduct an ablation study by training other single-codebook codecs (like X-codec2 or BiCodec) using the exact same pre-training, SFT, and LPO methodology?

---

### Official Review · Reviewer_rKPV · 2025-10-31

**Soundness:** 2
**Presentation:** 1
**Contribution:** 2
**Rating:** 2
**Confidence:** 4

**Summary:**

## Summary
 The paper proposes **UniTTS**, an end-to-end, LLM-based text-to-speech system that unifies text and audio token modeling. A key component is **DistilCodec**, a distilled neural audio codec that converts universal audio (speech, music, and sound effects) into a *single* large codebook of discrete tokens (32,768 entries) with high utilization, mitigating codebook collapse while keeping a single-stage decoding pipeline. UniTTS employs DistilCodec as audio tokens and pretrains with mixed audio/text objectives, then applies supervised fine-tuning and Linear Preference Optimization (LPO) to improve TTS performance. Experiments report high codebook usage (≈98.2%) and competitive subjective quality across fidelity, stability, naturalness, and emotional expressiveness; notably, the LPO-aligned model (**UniTTS-LPO**) improves over the SFT baseline on a diverse, author-constructed testset.


**Main contributions**

1. **DistilCodec**: a distillation method (DMS) that transfers a multi-codebook teacher to a *single-codebook* student, achieving high codebook utilization with 32,768 entries on universal audio, and aligning its 3,584-dimensional embeddings with the LLM to pass full acoustic detail into UniTTS.
2. **Unified, single-stage TTS with an LLM**: UniTTS jointly models audio and text token sequences (no separate semantic/acoustic stacks), simplifying decoding while preserving information compared to semantic-alignment approaches.
3. **Preference Alignment**: adoption of **LPO** for preference optimization in TTS, addressing repetition and prosody issues that persist after SFT and yielding consistent gains in subjective evaluation.
4. **Empirical validation**: measurements of codebook usage and audio reconstruction quality for DistilCodec, plus a comprehensive subjective TTS evaluation where **UniTTS-LPO** attains a higher average MOS than the SFT variant and is competitive with strong baselines (e.g., CosyVoice2, Spark-TTS).

**Strengths:**

**Clear and simple method design:** The work separates *perceptual* audio discretization (DistilCodec) from *cognitive* cross-modal generation (UniTTS), with a training pipeline—Pretraining → SFT → Alignment—that mirrors modern LLM practice.

**Improve single-codebook distillation:** With a 32,768-entry codebook, the paper reports ~98.2% utilization and provides objective comparisons against multiple codecs, supporting the claim that codebook collapse is mitigated.

**Unified modeling of audio signals:** During the training of the audio codec, a large-scale dataset was used to achieve a unified representation and modeling of diverse audio signals.

**Weaknesses:**

## Weakness

1. **Lack of fair comparison on TTS benchmarks:** The UniTTS evaluation metrics are mainly based on subjective listening tests. I believe the authors should conduct **fair comparisons on public test sets** (e.g., *seed-tts-eval* and *LibriSpeech PC test-clean*) and compare with current TTS baselines such as **SeedTTS, SparkTTS, CosyVoice, F5-TTS, and Index-TTS**, and report **objective metrics including WER and speaker similarity**. Additionally, for fair evaluation and reproducibility, the test set should be released so that reviewers can verify the reported results, and I think the author should train the model on public data for fair comparison.
2. **Lack of novelty**: I believe this paper shows **limited novelty**, as its overall approach is **quite similar to works such as SparkTTS** and CosyVoice, which also train large language models using **single-layer audio tokens**. The methodological overlap reduces the distinctiveness of the proposed contribution.
3. **Insufficient evaluation metrics for reconstruction tasks:** I suggest that the author should also report the **codebook utilization rate** for each codec in Table 3. As far as I know, **XCodec2** and **BiCodec**, which use single-layer codebooks, also maintain relatively high utilization rates. In addition, more **objective evaluation metrics**—such as **ViSQOL** and **FAD**—should be reported to better demonstrate the **generalization and universality** of DistilCodec across different types of audio.
4. **Code and data release:** Many data sources in this work are proprietary or self-constructed (e.g., the 100k-hour DistilCodec pretraining corpus and the comprehensive TTS evaluation set). The paper does not clearly state whether the **data and code will be released**, nor whether **model weights and inference scripts** will be made available. This lack of transparency negatively affects the **reproducibility** of the work.
5. **Performance not very strong:** On the *seedtts-eval* benchmark, as Table 6, UniTTS achieves a **CER of 3.466**, which is **not particularly strong** compared with existing state-of-the-art TTS systems.
6. **Potential issues in sample selection for Linear Preference Optimization (LPO):** In LPO, the real samples are used as **positive examples**, while the model-generated samples are treated as **negative examples**. However, in text-to-speech (TTS) tasks, the model-generated samples may sometimes sound **better than the real recordings**. Using the current method to assign positive and negative pairs could therefore **harm model performance**. I also suggest that the authors **report DPO (Direct Preference Optimization) results** as a fair comparison to evaluate the advantages and drawbacks of LPO more comprehensively.
7. **Minor writing and formatting issues:** There are occasional typographical errors and inconsistent notation throughout the paper. Careful proofreading and revision are recommended, along with improving the **visual quality and consistency of figures**.

**Questions:**

## Questions

1. Open-source: Will the UniTTS and DistilCodec be open source?
2. Fair comparison: Can we report fair results on public test sets and train the model on public data to prove its effectiveness？
3. universal modeling: It would be valuable to **verify DistilCodec’s performance on other types of audio generation tasks**, such as **general audio** and **music**, to further demonstrate the model’s **universality and general modeling capability** beyond speech reconstruction.

---

### Official Review · Reviewer_GSP8 · 2025-11-01

**Soundness:** 4
**Presentation:** 2
**Contribution:** 2
**Rating:** 2
**Confidence:** 3

**Summary:**

This paper presents UniTTS, an end-to-end Text-to-Speech (TTS) system based on a Large Language Model (LLM). The authors identify that a significant limitation in current LLM-based TTS systems is the imperfect alignment between semantic and acoustic information, which can be addressed by moving towards a more holistic modeling of audio. To this end, they propose a two-part contribution. First, they introduce DistilCodec, a novel neural audio codec that uses a distillation process to compress a multi-codebook model into a single, large-vocabulary (32,768 codes) model. This method is shown to effectively mitigate the codebook collapse problem that plagues large single-codebook models, achieving nearly 100% codebook utilization. Second, they build the UniTTS system, which uses DistilCodec as its audio tokenizer and employs a Qwen2.5-7B LLM to model the joint distribution of text and audio tokens. The system is trained using a comprehensive three-stage pipeline: pre-training on a large corpus of mixed data, supervised fine-tuning (SFT) on high-quality curated data, and alignment using Linear Preference Optimization (LPO). The authors conduct extensive experiments, showing that UniTTS achieves state-of-the-art results in speech quality, naturalness, and emotional expressiveness, outperforming several strong baseline models.

**Strengths:**

1. Novel and Effective Codec Distillation Method: The paper uses a distillation method to create a single-codebook codec. This method maintains high performance. The authors also verify that its codebook utilization is extremely high.
2. Comprehensive Experiments and Excellent Results: The paper evaluates both the codec and the full TTS system. The results demonstrate the effectiveness of the proposed methods.
3. Sufficient Training Details: The paper provides detailed information about the training process. This makes the work easier to reproduce.

**Weaknesses:**

1. Poor Writing Details: The writing has poor attention to detail. For example, the in-text citations do not handle parentheses correctly. It seems the authors did not distinguish between \citet and \citep in LaTeX. There is another error: the end of Section 3.2 refers to "Algorithm 3.2", which should be "Algorithm 1".
2. Over-packaging of the Method: The paper uses many formulas and pseudo-code for the DMS method. However, the method seems to only initialize the student's encoder and decoder with the teacher's weights. The rest of the student codec is trained from scratch. The paper does not explain the reason for choosing a codebook size of 32,768. It also does not analyze which part of the design leads to high codebook utilization.
3. Unfair Comparison of Training Data Size: In Section 4.3.1, the paper states its SFT stage uses much less data than LLaSA and Spark-TTS. However, this comparison is not fair. To my knowledge, LLaSA and Spark-TTS perform pre-training and SFT together. UniTTS should consider its pre-training data for a fair comparison. Table 16 shows the pre-training used 100B of audio data (about 300k hours).
4. Lack of Ablation Experiments on the Codec. The paper does not include ablation studies for the DistilCodec model itself.

**Questions:**

1. Could you please design ablation experiments for the codec and provide the reason why DistilCodec's codebook utilization is so high?
2. Could you provide more details about the MOS test? For example, how many raters were used and their background (e.g., native speakers, experts).
3. Regarding Weakness 3, I understand that pre-training can utilize data without paired text. However, the open-source datasets you list in Section 4.3.1 seem to have text transcriptions. Could you explain why you chose to use a pre-training stage instead of directly performing a large-scale SFT? In appendix C.10, you show that using 6.2 million text-audio pairs for SFT performs worse than UniTTS-SFT. However, I'm not sure what would happen if all the data from the pre-training were used for Pure-SFT.

---

### Official Review · Reviewer_WbC7 · 2025-11-01

**Soundness:** 2
**Presentation:** 2
**Contribution:** 2
**Rating:** 2
**Confidence:** 5

**Summary:**

The paper introduces UniTTS, a unified speech codec and text-to-speech (TTS) framework built around a single-codebook tokenizer distilled from a multi-codebook teacher, named DistilCodec. The approach aims to achieve high codebook utilization and compact audio tokenization, integrated with an LLM-based decoder and a preference alignment stage (LPO). While the motivation to merge codec and TTS modeling is clear, the framework mainly integrates existing components: codec distillation, autoregressive audio modeling, and alignment tuning into a single pipeline. Empirical results show reasonable but not superior performance to established codecs and TTS systems, and the necessity of the multi-to-single distillation step is not convincingly demonstrated.

**Strengths:**

1. The goal of combining codec and TTS modeling via a single high-utilization codebook (DistilCodec) is well motivated and practically useful for codec–LLM pipelines.

2. DistilCodec achieves high codebook utilization (98.2%), addressing the codebook collapse issue commonly observed in single-codebook tokenizers.

3. Integrating audio tokenization with LLM-based autoregression and a preference-alignment stage (LPO) is an interesting design that combines several recent ideas into a coherent TTS framework.

**Weaknesses:**

1. The necessity of distilling from a multi-codebook teacher to a single large codebook is unclear. A single large-codebook codec can be trained directly, as explored in prior work such as WavTokenizer, DAC, and XCodec2. The paper does not demonstrate that the multi to single distillation yields improvements beyond initialization convenience. Their DMS/DLF procedure is presented as necessary, but a direct single-codebook training baseline is missing.


2. DistilCodec's reconstruction quality is not clearly superior. In Table 3, DistilCodec shows lower STOI, PESQ, and UTMOS than established codecs (Encodec, DAC, BigCodec), calling into question whether the multi-stage distillation and large codebook materially improve audio quality.
Evaluations are missing standard reconstruction metrics and matched comparisons. Table 3 omits metrics commonly used in TTS/codecs evaluations (e.g., WER on reconstructed speech, ViSQOL, explicit similarity scores) and reports DistilCodec results primarily on LibriSpeech-Clean. A broader, matched-bitrate comparison (e.g., against DAC/WavTokenizer/XCodec2/BigCodec at the same token rates) is required to isolate the effect of distillation vs bitrate/configuration.


3. The distillation teacher choice is under-justified. The authors train a proprietary multi-codebook teacher and then performs knowledge distillation into the student DistilCodec model. But it would be more convincing to distill from strong, existing multi-codebook codecs (Encodec, SpeechTokenizer, etc.) to demonstrate the generality and advantage of DMS. The current pipeline risks being an engineering loop that simply compresses the model's own teacher.


4. UniTTS largely combines prior elements of codec-based tokenization (e.g., EnCodec, DAC), LLM autoregression for audio+text (e.g., Vall-E), and preference alignment into a single pipeline. The paper presents substantial work, but the methodical contribution beyond careful integration is limited.

5. Evaluation scope and robustness are narrow. DistilCodec reconstruction is evaluated mainly on LibriSpeech-Clean and a small universal-audio MOS study. There is little analysis on noisy, in-the-wild, multilingual, or low-resource scenarios despite claiming “universal audio” training. This weakens claims about generalization beyond the curated test sets.

6. LPO vs alternatives are unexplored. The Linear Preference Optimization stage is introduced as more stable than DPO, but the paper does not compare LPO against other alignment/optimization methods (DPO, reward modeling + RL, direct SFT with filtering) to justify this choice.

**Questions:**

1. Please see the Weakness section for points where additional analyses would strengthen the paper.

2. Can the authors provide controlled, bitrate-matched comparisons of DistilCodec with recent strong codecs (Encodec, DAC, BigCodec, StableCodec, XCodec2, WavTokenizer) on the same test sets, including WER and ViSQOL metrics?

3. What ablations show the added value of DMS/DLF vs directly training a single large codebook from scratch? Please report reconstruction and utilization trade-offs for a directly trained single-codebook baseline.

4. How robust is DistilCodec/UniTTS to noisy, conversational, or multilingual audio? Can the authors report quantitative reconstruction and MOS results on more diverse datasets beyond LibriSpeech-Clean?

5. Can the authors justify the choice of LPO through empirical comparison to DPO or other alignment methods?

---

### Note · Authors · 2025-12-03

I have read and agree with the venue's withdrawal policy on behalf of myself and my co-authors.